# Single-Cell Analysis of Primary Liver Cancer in Mouse Models

**DOI:** 10.3390/cells12030477

**Published:** 2023-02-01

**Authors:** Tina Suoangbaji, Vanilla Xin Zhang, Irene Oi-Lin Ng, Daniel Wai-Hung Ho

**Affiliations:** Department of Pathology and State Key Laboratory of Liver Research, The University of Hong Kong, Hong Kong SAR, China

**Keywords:** scRNA-seq, animal model, primary liver cancer

## Abstract

Primary liver cancer (PLC), consisting mainly of hepatocellular carcinoma and intrahepatic cholangiocarcinoma, is one of the major causes of cancer-related mortality worldwide. The curative therapy for PLC is surgical resection and liver transplantation, but most PLCs are inoperable at diagnosis. Even after surgery, there is a high rate of tumor recurrence. There is an unmet clinical need to discover more effective treatment options for advanced PLCs. Pre-clinical mouse models in PLC research have played a critical role in identifying key oncogenic drivers and signaling pathways in hepatocarcinogenesis. Furthermore, recent advances in single-cell RNA sequencing (scRNA-seq) have provided an unprecedented degree of resolution in such characterization. In this review, we will summarize the recent studies that utilized pre-clinical mouse models with the combination of scRNA-seq to provide an understanding of different aspects of PLC. We will focus particularly on the potentially actionable targets regarding the cellular and molecular components. We anticipate that the findings in mouse models could complement those in patients. With more defined etiological background, mouse models may provide valuable insights.

## 1. Introduction

Primary liver cancer (PLC) is the sixth most common cancer and the third leading cause of cancer-related deaths worldwide [1,2]. Geographically, the highest incidence areas are the East and Southeast Asia (e.g., China, South Korea and Japan) and sub-Saharan Africa [3]. Among all countries, China has the highest number (around 50%) of new cases of PLC worldwide [4] and 45% of PLC-related death [5,6]. Based on the histology, PLC consists mainly of hepatocellular carcinoma (HCC) and intrahepatic cholangiocarcinoma (ICC) [7] and HCC accounts for the majority (~80%) of PLCs [8]. HCC is a male predominant cancer, with a high male-to-female ratio ranging from 4–7 [9]. 

The key etiologies of HCC include chronic viral infection with hepatitis B or C virus (HBV or HCV), excessive alcohol consumption, non-alcoholic fatty liver disease (NAFLD), cirrhosis of any etiology, and exposure to carcinogens, such as aflatoxins [10,11,12]. Genetically, the accumulation of somatic genomic alterations, particularly of cancer driver genes, contributes to tumorigenesis [13]. Mechanistically, dysregulation of signaling pathways, including telomere maintenance, cell cycle control, oxidative stress, epigenetic and chromatin remodeling, and specific signaling such ass WNT-*β*-catenin and AKT-mTOR-MAPK signaling [2]. The tumor microenvironment also plays a significant role in promoting and maintaining the malignant transformation [14]. However, the mechanistic landscape of PLC is still far from completely understood.

Currently, the most effective curative treatment for early HCC patients is surgical resection and liver transplantation [15]. However, the majority of HCCs are diagnosed at advanced stages and therefore inoperable; even in tertiary medical centers, only about 20% of the tumors are operable [16]. Currently, the first-line drug treatment for advanced HCC is combined atezolizumab/bevacizumab targeting PD-L1/VEGF, while the alternative first-line drugs are the tyrosine kinase inhibitors (TKIs), sorafenib and lenvatinib [17]. However, the response rate to these systemic first-line drugs is about 20–30%, and hence the efficacy is still unsatisfactory [18,19]. There is an urgent unmet clinical need to find better treatment options and determinants for precision medicine for HCC.

ICC is the second commonest primary liver cancer. It has shown an increasing trend in the past decade [20]. For the large-duct type of ICC, the risk factors consist of intrahepatic biliary stones (hepatolithiasis), primary sclerosing cholangitis and liver biliary parasites, whereas for the small-duct type, several risk factors have been identified and include viral hepatitis, cirrhosis, obesity-associated liver disease, and diabetes [21]. However, about half of the ICC cases develop without identifiable risk factors [22], and most patients are diagnosed in advanced stages with a poor prognosis. Similar to HCC, surgical resection is the most effective way to treat ICC and there are very few alternative options available in other settings [23].

The PLC tumors are of high intratumoral heterogeneity, consisting of highly admixed cell populations residing in the tumor microenvironment. To this end, single-cell sequencing strategies are ideal for carrying out relevant cellular and molecular investigations. Regarding the development of single-cell RNA sequencing (scRNA-seq), Tang et al. first reported transcriptome sequencing analysis at single-cell level in 2009 [24]. Subsequently, more advanced methods were developed to detect the mRNA expression of genes at single-cell resolution, such as STRT-seq, Smart-seq2, CELseq2 and Drop-seq [25]. As a result, scRNA-seq has become a routine method to study the differences between transcriptomes of cells, revealing cell type composition, rare cell subtypes, cellular states, and developmental trajectories. As such, it has been widely used in the exploration of embryo and tumor development, pinpointing subpopulation identification, cell-cell interaction and lineage trajectory analyses.

Pre-clinical mouse models are particularly useful in recapitulating the process of hepatocarcinogenesis in human and have played a critical role in revealing the key oncogenic drivers and signaling pathways of liver cancer [26,27]. They are also frequently used to carry out studies on multiple stages of liver diseases [27], novel anti-cancer drug screening and development, and therapy testing [26,28]. Generally, the pre-clinical mouse models can be classified into three major subtypes: Diet-induced, chemical/carcinogen-induced, and genetic manipulation-induced spontaneous tumorigenesis [27]. Depending on the research objectives, some studies employ a combination of tumor-induction methods such as high-fat diet (HFD) combined with DNA-damaging carcinogen diethylnitrosamine (DEN) to recapitulate the carcinogenesis processes in human [29]. Combined with scRNA-seq, pre-clinical mouse models can provide a more comprehensive understanding of different aspects of liver cancer, including tumor initiation, growth and progression, and translational significance. In this review, we will summarize and compare the recent studies on pre-clinical mouse models that utilized the scRNA-seq to explore various research questions in liver cancer. 

## 2. Insights Derived from Animal Models of PLC

### 2.1. Diet-Induced Model

The positive correlation between NAFLD or nonalcoholic steatohepatitis (NASH) and the risk of HCC is well established. NAFLD is a common syndrome in people who are overweight or obese, whereas NASH is a combination of diabetes, hypertension and obesity. NAFLD is the most prevalent etiology of HCC in western countries due to the growing obesity epidemic. The incidence of HCC among patients with NASH-associated cirrhosis is approximately 2.6% per year [30]. To mimic this obesity-associated lifestyle, HFD-induced or western-style diet (WD)-induced NAFLD-HCC mouse models are well-established. These diets usually contain more than 40% butter fat and high glucose and high cholesterol [31]. Choline-deficient HFD (CD-HFD) is a commonly used diet to induce NAFLD-related liver diseases. Choline and methionine are the essential amino acids for very low-density lipoprotein synthesis and fatty acid *β*-oxidation in mitochondria [32]. Therefore, CD-HFD interrupts the metabolism of fatty acids, causes abnormal deposition of TG in liver, and results in NAFLD-associated liver damage [33]. Previous studies showed that C57BL/6J mice developed HCC at week 36 after feeding with HFD that is deficient in choline and L-amino acid [33]. A recent study has uncovered that CXCR6+ CD8 T cells were activated and triggered auto-aggression resulting in damage of the livers in mice fed with CD-HFD [34]. A comprehensive study [35] revealed that NASH-associated macrophages, marked by Trem2 expression, were the key determinants for disease severity of steatohepatitis. Moreover, crosstalks between hepatic stellate cells and other cells via respective secreted factors (as known as stellakines) and receptors also served as unique components for the NASH pathogenesis.

Notably, Pfister et al. utilized the scRNA-seq analysis on pre-clinical NASH-associated HCC mouse models and demonstrated that immune checkpoint blockade (ICB) therapy targeting programmed death-1 (PD1) indeed exacerbated, rather than suppressed, NASH-HCC, unlike HCC of other etiology such as HBV infection [36]. Firstly, they established NASH-HCC models with CD-HFD and western-style diet with trans-fat (WD-HTF) in C57BL/6 mice for 3–13 months to cause progressive liver injuries which can mimic the NASH-HCC in human. Then scRNA-seq was performed in the liver tissues from mice fed with normal diet (ND) and CD-FHD at 12 months. A significant enrichment and activation of hepatic CD8^+^PD1^+^ T cells related to cytotoxicity and effector-function was observed in the livers of mice with NASH using scRNA-seq on cells expressing T cell receptor *β*-chains (TCR*β*). Moreover, this type of cell expressed genes that are related to cytotoxicity and effector functions (Gzmk and Gzmn), as well as some exhaustion markers (Pdcd1 and Tox). While anti-PD1 treatment is an effective therapy in non-NASH HCC to a certain degree, it did not result in significant regression of tumors in the NASH-HCC mice. Besides, they also displayed increased fibrosis and contained an increased number of CD8^+^PD1^+^ T cells, which indicates that CD8^+^PD1^+^ T cells failed to execute an effective immune surveillance role, and instead showed tissue-damaging potential. They validated their hypothesis by depleting CD8^+^ T cells in NASH mice without liver cancer and found that CD8^+^PD1^+^ T cells not only had defected immune surveillance but were also able to promote HCC development in a NASH background. Consistently, similar results were also found in patients with NASH.

### 2.2. Chemical-Induced Models

The gradual accumulation of genetic alterations in the genome occurs during the onset of PLC. To this end, in order to recapitulate the processes in mice, toxins and carcinogens are administrated in mouse models. There are some highly reproducible and frequently used chemicals that can cause chronic or acute liver damage, including DEN, carbon tetrachloride (CCl4), streptozotocin (STZ) and alcohol. DEN can directly form covalent bonds with DNA leading to oncogenic mutations and immune modifications that are associated with HCC initiation and progression [37]. A single dose of DEN is effective enough to induce hepatocarcinogenesis in mice [38]. Hepatotoxin CCl4 can induce inflammation, oxidative stress and fibrosis in the liver and is widely used in animal models with acute liver injury [39]. STZ is an alkylating antineoplastic agent and is toxic to insulin-producing beta cells [40]. A low dose of STZ can induce hepatic steatosis and diabetes in neonatal mice. Hence, HFD administration in STZ-injected mice (STZ-HFD model) results in inflammation, NASH, fibrosis and HCC in the liver [41].

A recent study by Zhang et al. utilized chemical-induced an HCC mouse model and scRNA-seq to uncover the tumor suppressor role of the zinc finger protein Miz1 [42]. In this study, when Miz1 (*Miz*1^∆*hep*^) or its essential transcriptional POZ domain (*POZ*^∆*hep*^) was specifically deleted in hepatocytes, a 2-fold increase of tumor mass was observed when compared with wild-type (WT) littermates (F/F) control mice in DEN/CCl4-induced and STZ-HFD induced HCC models. ScRNA-seq was conducted in the tumor tissues of mice in the DEN/CCl4 model. While there was no significant difference in the cell populations between (*POZ*^∆*hep*^) and *Miz*1*^F/F^* mice, an interesting sub-cluster (sc-2) was identified in the pool of hepatocytes using both *Miz*1*^F/F^* and *Miz*1^∆*hep*^ scRNA-seq data. According to the Kyoto Encyclopedia of Genes and Genomes (KEGG) database, this sub-cluster demonstrated the highest activation level of the NF-kB pathway. This sub-cluster was also only highly enriched in the *Miz*1^∆*hep*^ models. Moreover, such hepatocytes showed a special cell state with high expression levels of inflammatory cytokines and chemokines, such as Tnf, Cxcl2, and Ccl4. These inflammatory cytokines and chemokines were able to promote the polarization of tumor-infiltrating macrophages towards pro-inflammatory activity. The study also validated that, mechanistically, the loss of Miz1 could cause NF-kB activation. Previous papers reported that the major NF-kB transactivating subunit RelA, when bound to oncoprotein metadherin (MTDH), activated the NF-kB signaling pathway upon stimulation by tumor necrosis factor *α* (TNF-*α*) [43,44]. Notably, it was verified that Miz1, through the sequestration of oncoprotein MTDH, inhibited the activation of the NF-kB pathway. More importantly, the expression of hepatocyte Miz1 in patients’ HCC was negatively correlated with phosphorylation of RelA and MTDH, as well as worse overall survival and higher recurrence rates. Taken together, loss of Miz1 in hepatocytes may generate a unique sub-lineage of hepatocytes with inflammatory features in the HCC mouse model and Miz1 exerted a tumor suppressor role in HCC by restricting hepatocyte-driven macrophage activation and inflammation.

### 2.3. Genetic Models

#### 2.3.1. Pure Genetic-Based Models

To better reflect the clinical complexities due to intra- and inter-tumor heterogeneity, transgenic mice can be engineered with the expression of oncogenes and/or constitutive or conditional silencing of tumor-suppressor genes [27]. Nowadays, retroviral infection, DNA microinjection into the pronucleus of fertilized oocyte, gene- and organ-targeted transgene approach via manipulating selected loci in mouse embryonic stem (ES) cells, and CRISPR/Cas9-based hydrodynamic tail vein injection (HDTVi) [45] are popular methods to generate transgenic mouse models [46]. 

In PLC, phosphatase and tensin homolog (PTEN) is a well-known tumor suppressor [47]. PTEN mutations are observed in many cancer types including HCC and ICC, and they are critical for liver tumor initiation and progression [48]. Hepatocyte-specific PTEN-knockout (KO) mice underwent tumorigenesis from hepatic steatosis, NASH to HCC, a process that is similar in human [48]. Therefore, this ‘loss of function’ transgenic mouse model serves as a useful pre-clinical tool to investigate the pathogenesis and mechanisms for PLC. AT-Rich Interaction Domain 1A (ARIA1A) is another frequently mutated tumor suppressor in HCC. There was a recent study [49] that indicated that scRNA-seq of hepatocytes isolated from liver-specific ARID1A-KO mice demonstrated an enhancement in stem/progenitor cell features, including cell differentiation state and cancer stem cell marker expression. Mechanistically, loss of ARID1A dysregulated gene expression machinery related to stem/progenitor cell maintenance, cell proliferation and self-renewal of hepatocytes. As ARID1A is a key component of SWI/SNF chromatin remodeling complex, the gene expression consequences were suggested to be dysregulated via alterations in chromatin accessibility.

Interestingly, a recent study performed sleeping beauty (SB) transposon-based screening in liver-specific PTEN-KO ICC mouse model and identified Traf3 to be the most significant trunk driver for tumor development [50]. Next, they generated hepatocyte- and cholangiocyte-specific Traf3/Pten double KO (HDKO and CDKO) mice and conducted lineage tracing using Rosa26-LacZ reporter. Importantly, keratin-19 (KRT19)-positive cholangiocytes and ICC were identified with LacZ staining in the HDKO mice, suggesting that ICC likely originates from Traf3/Pten-deficient hepatocytes. Therefore, to better understand the development of ICC, which originated from both Traf3 and Pten deficient hepatocytes, the researchers performed scRNA-seq using liver cells obtained from the HDKO mice and control wildtype (WT) mice. They analyzed about 10,000 cells, including hepatocytes, endothelial cells, macrophages, dendritic cells, monocytes, T cells, and B cells. Furthermore, the trajectory analysis for these single-cells revealed that a special cell population that expressed both typical hepatocytic marker Alb and typical cholangiocytic marker Sox9 transdifferentiated from normal hepatocytes in the HDKO mice. The inhibition of Traf3 and Pten promoted the transdifferentiation of hepatocytes to cholangiocytes. Subsequently, they validated the transdifferentiation phenotype in vitro by co-inhibiting TRAF3 and PTEN in HepG2 cells, which resulted in down-regulation of hepatocyte markers and up-regulation of cholangiocyte markers. The findings were further confirmed using murine liver organoids. Taken together, these experimental analyses confirmed the potential transdifferentiation capacity of hepatocytes into cholangiocytes. In summary, using transgenic mouse models and scRNA-seq, TRAF3 was identified as a potential tumor suppressor and targeting the TRAF3-NIK axis may provide a potential therapy for ICC.

Some studies employed SB transposase-based HDTVi spontaneous ICC tumorigenesis model with overexpression of AKT and NICD (intracellular domain of the NOTCH1 receptor) in mouse hepatocytes to investigate the transcriptomic profiles of PLC [51,52]. The persistent activation of AKT leads to an increase of cell proliferation and reduction of apoptosis, whereas the overexpression of NICD can activate NOTCH signaling in the liver and enhance biliary differentiation finally resulting in ICC tumorigenesis [53]. Using AKT/NICD transgenic model combined with scRNA-seq, Wang et al. detected some important roles of stromal cells in ICC [54]. Firstly, the authors performed scRNA-seq using ICC tissues from AKT/Notch intracellular domain–induced mouse model. They selected the ICC tissues at days 10, 17, and 31, which were regarded as early, middle and late stages of ICC development, respectively. After the quality control, there were 51,897 cells from ten cell types, including 16,344 T cells, 6221 macrophages, 2667 endothelial cells, and 26,665 other cells. They also compared the cell type distribution in mice with another human ICC scRNA-seq data and identified a consistent pattern. Next, to investigate the cellular heterogeneity in the ICC tumor microenvironment (TME), they extracted epithelial cells from all stages and re-defined eight epithelial subgroups in terms of their characteristic gene-expression patterns. They also confirmed the malignant ICC cells according to the analysis of copy number variation (CNV). Because of the high expression level of stress-responding genes Jun and Fos, and extracellular stimuli-responding genes Fgfr2 and Igf1r, a subpopulation of epithelial cells (AP1-C) was defined as the stress-responding subtype. Likewise, another subtype, which showed high expression levels of cell proliferation markers, such as Mki67 and Cdk1, was referred as the proliferating subtype (Mki67-C). Based on the weighted correlation network analysis (WGCNA), least absolute shrinkage and selection operator (LASSO), and hypergeometric test, they identified the specific transcription factors (TFs) in these two special subtypes (AP1-C and Mki67-C). The results indicated that Zmiz1 and Ybx1 were the core modulators in the AP1-C stress-responding and Mki67-C proliferating subtypes, respectively. Finally, the study also revealed that endothelial cells and fibroblasts interacted with each other and adapted themselves to promote ICC development. Four and five subtypes of tumor endothelial cells and fibroblasts, respectively, were identified. Interestingly, both these subtypes in the early and middle stages of ICC showed pro-inflammatory function, whereas the subtypes in the late stage of ICC established a tumor-promoting role. Notably, in the late tumor stage, through TGF-*β* and calcium signaling, one subtype of tumor endothelial cells (TEC- NOTCH) promoted the differentiation of fibroblasts into myofibroblastic CAF (myCAF) subtype and aggravated the ICC growth. Hence, the findings revealed functionally distinct cell subpopulations within ICC and provided hints to better understand ICC initiation and progression.

Besides overexpressing more than one oncogene, the combination of overexpressing an oncogene and depleting a tumor suppressive gene is another commonly used strategy in HDTVi spontaneous tumorigenesis models. *KRAS^G^*^12*D*^/p19 and YAP/AKT are popular combos for ICC preclinical mouse models. *KRAS^G^*^12*D*^ is a common oncogenic mutation in ICC patients and is associated with overgrowth of tumor cells [54]. Tumor suppressor p19 can stabilize P53 via inhibiting MDM2-mediated protein degradation [55]. YAP (Yes-associated protein) is one of the negative regulators of AKT and exerts either tumor suppressive or oncogenic roles in various types of cancer [56,57]. It was reported that YAP enhanced cancer stem cell (CSC) properties in HCC and ICC [58,59]. The emerging research in cancer-associated fibroblast (CAF) highlights the essential roles of CAFs in cancer development [60]. These cells promote cancer metastasis through angiogenesis, extracellular matrix (ECM) remodeling, and active immunosuppression [61] and play an integral part in the TME [62]. Around 90% of fibroblasts in the liver originate from hepatic stellate cells (HSC) [63]. An impressive study by Affo et al. revealed the diverse roles of CAF subpopulations in HDTVi ICC mouse models and confirmed that CAFs derived from HSCs could enhance ICC progression by directly interacting with tumor cells [64]. Totally, four murine ICC samples and six human ICC samples were sequenced and discovered the predominant cell types in the liver, such as hepatocytes, endothelial cells, HSCs, and CAFs. After annotating CAFs in both human and mouse data, they obtained three subpopulations of CAFs, namely HSC-CAFs, portal fibroblast (PF)-CAFs, and other CAFs. HSC-CAF contributed around 90% of the total CAFs in both of the two datasets. Meanwhile, the results of cell–cell interaction from CellphoneDB analysis showed that the majority of CAFs in murine ICC have strong communication levels with tumor cells in comparison with other cell types, which was also consistent with human ICC and pancreatic ductal adenocarcinoma (PDAC) scRNA-seq data. They suggested HSC-derived CAFs could promote ICC growth and uncovered subpopulations of CAFs, namely inflammatory and growth factor-enriched CAF (iCAF), myofibroblastic CAF (myCAF) and mesothelial CAF (mesCAFs). Of note, iCAFs expressed high levels of quiescence markers (Lrat, Reln and Rgs5) but a low level of activation markers (Col1a1, Acta2, Col8a1, Col15a1, Crlf1 and Fbn2). They were also enriched for specific pathways. myCAFs expressed lower quiescence and higher activation markers than iCAFs and enriched for ECM pathways. Additionally, both myCAFs and iCAFs exhibited a strong interaction level with malignant cells. The differently expressed genes were examined in CAF subpopulations. Has2 and Hgf were the most significantly upregulated genes in myCAFs and iCAFs, respectively. In terms of ligand-receptor analysis, the researchers also detected Ddr2 to be the receptor of Has2, whereas Met to be the receptor of Hgf. Importantly, the two receptor genes were highly expressed in the tumor cells. In summary, using combined scRNA-Seq and transgenic mouse models with linage tracing of CAFs, the studies uncovered the distinct CAF clusters involved in ICC growth, which may be targetable for treatment.

Up-regulation of c-Myc was found in 50% of various types of human cancers [65]. Abnormal amplification of c-Myc is one of the leading genetic alterations in hepatocarcinogenesis [65]. Overexpression of c-Myc, together with the co-expression of another oncogene or simultaneous deletion of a tumor suppressor, in hepatocytes can trigger spontaneous tumor initiation and progression in mice. Hydrodynamic transfection of oncogene c-Myc is a well-established mouse model of HCC [66]. However, the mechanism of this pro-oncogenic process is not well elaborated. Interestingly, a study combining HDTVi and scRNA-seq demonstrated the bi-directional roles of Shp2 in HCC [67]. In this study by Chen et al., c-Myc was transfected in WT and hepatocyte-specific Shp2 knockout (SKO) mice and revealed that Myc alone did not induce liver tumors, whereas Myc transfection in SKO mice induced tumor development. Next, the researchers performed scRNA-seq for WT and SKO mice at day 0 before Myc transfection (D0-Myc), day 10 (D10-Myc) and four weeks (W4-Myc) after transfection. They obtained 27,327 cells, including hepatocytes, tumor cells and other major cell types. Hepatocytes in W4-Myc SKO mice were isolated and re-clustered into five subgroups. Notably, one of the populations was tumor cells with highly expressed Afp and Myc, which were then annotated as Myc+ tumor cells. Interestingly, this subpopulation showed a high expression level of Shp2 as compared with normal hepatocytes, and this indicated the requirement of Shp2 and Myc to induce tumorigenesis. However, transfecting both Myc and Sph2 was inefficient to induce tumor but exacerbated the tumor growth in SKO livers. By using Alb-Cre- and AAV-Cre-induced Shp2 deletion that showed similarly effective Shp2 deletion, AAV-Cre induced acute, synchronized gene deletion whereas Alb-Cre mediated progressive gene ablation. Tumor induced by Myc required cell-autonomous expression of Shp2 but also required a hepatic microenvironment induced by Shp2 loss in the majority of hepatocytes. These data suggest that Shp2 acts like a double-edged sword in Myc-driven HCC. Moreover, they also revealed the functional requirement of *β*-catenin for Myc-driven hepatocarcinogenesis, providing an explanation for the frequent co-occurrence of aberrant activation of Myc and *β*-catenin signaling in human HCC.

#### 2.3.2. Inducible and Reporter Models

To investigate specific gene functions at transcriptional or translational levels, reporter knock-in mouse models are well-established. Fluorescence proteins and bioluminescence (including Green fluorescent protein (GFP), red fluorescent protein (RFP) and firefly luciferase) are conventional markers used in reporter knock-in transgenic mouse models [46]. When a target gene is expressed in mice, it can be visualized by the reporter’s fluorescent signals in situ. In vivo tumor growth, cell lineage tracing and live cell imaging can be monitored in mice carrying GFP/RFP or firefly-luciferase labeled tumor cells. There are several common reporter transgenic mouse lines: random reporter lines, Cre reporter and ROSA26 reporter lines [68]. A ROSA26 reporter strain is the most popular one. It contains various inserted reporter genes, a promoter sequence, a splice acceptor sequence (SA), and an expression element flanked by two loxP sites with the same direction as the Rosa26 locus [68]. Therefore, Cre recombinase activity on the specific gene target can be monitored at Rosa26 locus at any desired time point via reporter signals [46].

A recent study used knock-in reporter mouse line, carcinogen-induced HCC model and scRNA-seq to conduct cell linage tracing in mouse and has revealed a subpopulation of Prom1-expressing HCC cells with CSC properties [69]. It showed that the expression level of Prom1 was increased in both reporter knock-in mouse HCC model and human HCC data from The Cancer Genome Atlas (TCGA). Moreover, the proportion of Prom1^+^ HCC cells increased with HCC development. Next, they tried to deplete Prom1^+^ HCC lineage in vivo and observed that HCC growth and malignant progression were impeded, which indicates that Prom1^+^ HCC cells have CSC-like features. They further performed scRNA-seq for Prom1^+^ and Prom1^-^ models at different time points (days 3, 10, and 30). Almost a total of 34,000 cells were obtained. In the Prom1^+^ cells, the dominant cell type was HCC cells, accounting for about 95% throughout the disease development, suggesting that the Prom1^+^ cells did not transdifferentiate into other lineages in this chronic HCC model and they were the expanding epithelial tumor cells during HCC progression. They detected well-known CSC markers expressed in Prom1^+^ cells with distinct pattern, suggesting the heterogeneity of Prom1^-^derived HCC subpopulations. Functional annotation analysis also indicated Prom1^+^ cells lost most of the normal liver functions, which may represent dedifferentiation status. More importantly, Prom1-lineage gene signature predicted poor prognosis HCC patients and the malignancy might depend on the activated oxidant detoxification system. In summary, the combination of in vivo cell linage tracing and scRNA-seq has uncovered a unique CSC-like pro-oncogenic role of Prom1^+^ HCC cells and this finding may shed light on the HCC cell heterogeneity in human HCC.

### 2.4. Liver Cancer Xenograft Models

Patient-derived tumor xenograft (PDX) mouse models are a commonly employed translational research tool to facilitate precision medicine. Resected clinical tumor tissues are transplanted into immunodeficient mice [70]. PDX mouse models can maintain the biological and pathological characteristics of patients’ tumors and recapitulate the oncogenic TME [71]. In the TME, stromal cells, such as endothelial cells, immune cells, tumor-associated macrophages (TAMs) and CAFs, are surrounded by ECM consisting of a mixture of proteins and proteoglycans [9]. Therefore, humanized PDX mouse models are widely utilized in cancer research, including anti-cancer drug development and screening, co-clinical trials, ICB therapy testing, personalized medicine and establishment of PDX biobank [70]. Our previous study revealed the frequent TSC mutations in HBV-associated HCC. By using relevant TSC-mutant PDXs, we demonstrated the hypersensitivity of precision treatment towards rapamycin, thus pinpointing mutation-dependent mTOR hyperactivation [72]. Moreover, we also explored the intra-tumoral heterogeneity and identified stemness-related subpopulations in HCC using PDX model [73,74].

In order to depict the primary resistance mechanism of sorafenib, the first-line kinase inhibitor drug for HCC, the study by Guan et al. [75] inoculated liver tumor tissues from three patients into mouse livers to establish PDXs and performed scRNA-seq analysis. As evaluated by the survival rates after treatment with sorafenib, they defined a group of PDX model as sorafenib primary resistance (PR) with low survival rates, and two other groups with relatively higher survival rates as sorafenib sensitive A (SA) and sorafenib sensitive B (SB). The scRNA-seq data of PR group were compared with that of the other two groups (SA and SB). The data suggested the potential role of liver bud hepatic cells in primary resistance to sorafenib and the regulation by the JUN transcription factor, hypoxia HIF signaling and neonatal Fc receptor (FcRn) activation.

## 3. Discussion and Future Directions

In recent studies [35,36,42,50,52,64,67,69,75], scRNA-seq data analyses on pre-clinical mouse PLC models have been made use of to investigate the cellular composition, cell crosstalk, CNV inference, and trajectory analysis (Figure 1). Most of them have focused on the hepatocytes and several non-parenchymal cell types from PLC tissue (Table 1).

By integrating the findings of the aforementioned papers, we have noted some key insights: one is the pivotal involvement of NF-kB signaling in PLC, and the other is the functional significance of CAF, which might be worthy for further functional characterization as therapeutic targets. Regarding NF-kB signaling, NF-kB-inducing kinase (NIK) demonstrated an inverse correlation with TRAF3 (a potential tumor suppressor in ICC), and the targeting of TRAF3/NIK axis may provide potential therapy in ICC patients [48]. Moreover, NF-kB inhibitors have also been developed and tested in pre-clinical models. To this end, some studies have evaluated the in vitro effect of Dehydroxy-methyl-epoxyquinomicin (DHMEQ) as an NF-kB inhibitor in human HCC cell lines [76,77], but there are still challenges to overcome in making NF-kB inhibition clinically applicable.

However, CAF is one of the most abundant cell types in the TME and it supports multiple aspects of tumor development by suppressing antitumor immune responses and remodeling the TME to support tumor growth. Some studies [52,64] have also paid attention to the function of CAF in PLC. They demonstrated that in ICC, most CAFs were derived from HSCs, which could facilitate ICC growth, and the cell-cell communication between CAF and endothelial cells also enhanced the ICC development. In other study [78], CAF could act as a tumor-supportive cell type to create an advantageous microenvironment for ICC aggressiveness and chemotherapy resistance by educating the stemness-enhancing capacity of myeloid-derived suppressor cells (MDSC). Moreover, our previous study [79] also observed the presence of CAF positively correlated to poor clinical outcome. In fact, there are on-going studies investigating the immunotherapeutic strategies that target CAF in cancers, including HCC [80].

Many studies focused on the immune components of the TME of PLC tumors. Notably, there were recent and insightful findings utilizing lineage tracing and mouse models to pinpoint the importance of CSC marker-expressing cells in the tumor initiation and progression [69]. We believe the use of relevant mouse models can effectively recapitulate the genuine hepatocarcinogenesis process and provide valuable exemplification for the future delineation of CSCs in PLC.

In summary, using animal models to pre-clinically study PLC can help deepen our understanding of the pathogenesis of PLC. The pre-clinical mouse models are extremely valuable in recapitulating the process of hepatocarcinogenesis [81,82]. By applying scRNA-seq technology in these models, we can discover and analyze rare cell populations in the tumors and investigate the heterogeneity of the TME. 

Recently, more attention has been paid to utilize spatial transcriptomics (ST) in studying HCC, which can provide the spatial information of liver zonation and other applications [83]. Compared with scRNA-seq, ST can preserve the valuable spatial information and at the same time reveal the cellular and molecular profiling information at nearly single-cell resolution. This is an attractive merit that favors the adoption of ST in the investigation of human cancers [84], for which the in-situ information is essential for delineating intra-tumoral heterogeneity and tissue architecture. We anticipate that there will be emerging studies in PLC by making use of spatial transcriptomics coupled with scRNA-seq. In fact, we noted that there are some recent reports utilizing ST to reconstruct the spatial architecture of PLC tumors and other liver diseases [85,86,87].

**Table 1 cells-12-00477-t001:** Summary of the studies using PLC or NASH pre-clinical mouse models combined with scRNA-seq analysis.

Reference	Type of PLC	Mouse Model	Platform	No. of Cells	Cell Type	Sample	Key Findings	Data Accession Number
[52]	ICC	Genetic model (overexpression of AKT/NICD)	10x	Total: 51,897 Endothelial: 2667 Fibroblast: 1442	Ten cell types, focused on stromal cells	Tumor tissue on 7, 13 and 31 days and whole normal liver tissue	Reveal a stress-responding and proliferation subtypes of cells in ICC which can be druggable	PRJNA743579
[50]	ICC	Genetic model (Traf3/Pten double KO)	BD Rhapsody	10,364	Seven cell types, focused on hepatocytes	Liver tissue with tumor and normal liver tissue	TRAF3 acts as a potential tumor suppressor in ICC	GSE178814
[64]	ICC	Genetic model (overexpression of YAP/AKT and depletion of KRAS/p19)	10x	N/A	CAF-enriched but diverse cell types	Tumor issue	Most CAFs are derived from HSCs, and inflammatory CAFs promote ICC growth	GSE154170
[69]	HCC	Inducible and reporter model (Overexpression of DEN/CCL_4_ and knock-in Rosa26-LSL-Tomato reporter)	10x	Total: 33,473 HCC: 2202	Eight cell types, focused on HCC cells	Tumor tissue on day 3, 10 and 30	Prom1^+^ cells enhanced HCC proliferation and progression with CSC-like properties	GSE181515
[42]	HCC	Chemical-induced model (DEN/CCL_4_ and STZ-HFD)	10x	Total: 24,802 Hepatocyte: 2365 Macrophage: 4277	Seven cell types, focused on hepatocytes and macrophages	Tumor tissue	Miz1 acts as a tumor suppressor in HCC	GSE142868
[36]	HCC	Diet-induced model (CD-HFD and WD-HTF)	MARS-Seq	N/A	Leukocytes, particularly T cells	Liver tissue with NASH and normal liver tissue	Potentially unique pro-cancer role of CD8^+^PD1^+^ T cells in NASH-HCC	GSE144635
[67]	HCC	Genetic model (Myc transfection and Shp2 KO)	10x	27,327	Focused on hepatocytes but with other non-parenchymal cells	Liver tissue at different time points	Bi-directional role of Myc in suppressing or promoting HCC	GSE157561
[75]	HCC	PDX model	BD Rhapsody	10,602	All cell types	Liver cancer tissue	A specific cell cluster of liver bud hepatic cells showed a sorafenib-resistance role	GSE175716
[35]	HCC	Diet-induced model (AMLN)	10x	33,168	Focused on NPC	Healthy and NASH liver tissue	A new subtype of macrophages called NAMs presented features of NASH and was associated with disease severity	GSE129516

CAF, cancer-associated fibroblast; CCL_4_, carbon tetrachloride; CD-HFD, Choline-deficient-high-fat diet; CSC, cancer stem cell; DNE, diethylnitrosamine; HCC, hepatocellular carcinoma; HSC, hepatic stellate cells; ICC, intrahepatic cholangiocarcinoma; KO, knockout; MARS-seq, massively parallel RNA single-cell sequencing; NAM, NASH-associated macrophages; NPC, non-parenchymal cell; PDX, Patient-derived tumor xenograft; PLC, primary liver cancer; STZ, streptozotocin; WD-HTF, western-style diet with trans-fat.

## Figures and Tables

**Figure 1 cells-12-00477-f001:**
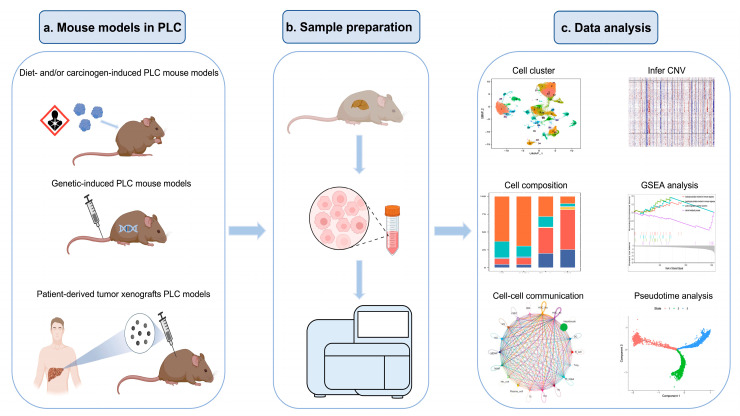
Overview of the workflow of scRNA-seq analysis of PLC in mouse models.

## Data Availability

The data presented in this study are openly available.

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
