# Peer review of "Single-Cell Analysis of Primary Liver Cancer in Mouse Models"

_cells, 2023, doi:10.3390/cells12030477_

Round 1

Reviewer 1 Report

In the review entitled with “Single-cell Analysis of Primary Liver Cancer in Mouse Models”, Suoangbaji et al describe the recent advances in single-cell RNA sequencing (scRNA-seq) in mouse liver cancers, which provide some interesting and valuable information, and even insights to pathogenesis of liver cancers. 

However, there are some concerns, as following:

1.     The literatures related to the field are incomplete, some papers are missed, including the scRNA analysis on livers of Arid1a knockout mice or others.

2.     In Discussion and Future Directions, the authors only mentioned two insights, which is too simple. As known, scRNA-seq has become a routine method to reveal cell type composition, rare cell subtypes, cellular states, and developmental trajectories. In addition to NF-kB signaling and CAFs, the authors may provide more insights into initiation and progression of liver cancers, or point out the future direction in detail, based on the scRNA-seq or snRNA-seq.

3.     Table 1 also is too simple, and the author may describe mouse models (which genes were genetically inserted or knocked out), methods (10X or others) of scRNA or snRNA, and major findings in cell types, in more detail.

4.     If possible, the authors may summarize some important insights to pathogenesis of liver cancers, based on the scRNA or snRNA data, by combining with known knowledge.

Author Response

Thank you for your effort in reviewing the manuscript. We found the comments are constructive and useful. Our response is as follows:

Comment 1: The literatures related to the field are incomplete, some papers are missed, including the scRNA analysis on livers of Arid1a knockout mice or others.

Response: We have added the suggested study together with others into our manuscript.

Comment 2: In Discussion and Future Directions, the authors only mentioned two insights, which is too simple. As known, scRNA-seq has become a routine method to reveal cell type composition, rare cell subtypes, cellular states, and developmental trajectories. In addition to NF-kB signalling and CAFs, the authors may provide more insights into initiation and progression of liver cancers, or point out the future direction in detail, based on the scRNA-seq or snRNA-seq.

Response: We have added the discussion on the role of CSCs in tumor initiation and the use of spatial transcriptomics for future studies.

Comment 3: Table 1 also is too simple, and the author may describe mouse models (which genes were genetically inserted or knocked out), methods (10X or others) of scRNA or snRNA, and major findings in cell types, in more detail.

Response: We have added the suggested details, including description of mouse models, scRNA-seq platform and key findings.

Comment 4: If possible, the authors may summarize some important insights to pathogenesis of liver cancers, based on the scRNA or snRNA data, by combining with known knowledge.

Response: Our review focuses on the use of single-cell sequencing and mouse models to investigate primary liver cancer. We have now highlighted some key insights regarding the signalling pathway, CAF and CSC in the discussion. We also pointed out that spatial transcriptomics technology may be the future direction of investigation.

Reviewer 2 Report

In the review titled “Single-cell analysis of primary liver cancer in mouse models,” by Suoangbaji & Zhang et al., the authors nicely describe several different mouse models and key findings that have been discovered using single cell sequencing technology on tissue from these models. The sections are well organized by the method used to induce carcinogenesis. With the addition of a summary figure and some small edits, I believe this review will be a useful resource for the research community.

For a review, I feel the authors provide too many details on few studies. I recommend including summaries for 2-4 publications for each type of model (diet-induced, chemical-induced, genetic and xenograft) and keeping the text focused mainly on the single cell data from each study. Study summaries should be shortened to the results obtained based on the single cell sequencing. For example, lines 131-142 could be removed. With limited publications on NASH-HCC in mouse models, this paper would be interesting to include instead that profiles NASH in a mouse model at the single cell level: “Landscape of Intercellular Crosstalk in Healthy and NASH Liver Revealed by Single-Cell Secretome Gene Analysis,” by Xiong et al., Molecular Cell, 2019. Lines 214-231 can also be removed. Lines 258-264 can be removed.

Please include a figure summarizing key findings such as stem-like characteristics of tumor cells, microenvironment composition, lineage tracing and unique cell type populations.

Changing the title of 2.3.2 to inducible and reporter models. Genetic-diet may get confused with the diet induced systems presented earlier in the review.

Author Response

Thank you for your effort in reviewing the manuscript. We found the comments are constructive and useful. Our response is as follows:

Comment 1: For a review, I feel the authors provide too many details on few studies. I recommend including summaries for 2-4 publications for each type of model (diet-induced, chemical-induced, genetic and xenograft) and keeping the text focused mainly on the single cell data from each study. Study summaries should be shortened to the results obtained based on the single cell sequencing. For example, lines 131-142 could be removed. With limited publications on NASH-HCC in mouse models, this paper would be interesting to include instead that profiles NASH in a mouse model at the single cell level: “Landscape of Intercellular Crosstalk in Healthy and NASH Liver Revealed by Single-Cell Secretome Gene Analysis,” by Xiong et al., Molecular Cell, 2019. Lines 214-231 can also be removed. Lines 258-264 can be removed.

Response: We have added the suggested study and trimmed the text according to the suggestions. 

Comment 2: Please include a figure summarizing key findings such as stem-like characteristics of tumor cells, microenvironment composition, lineage tracing and unique cell type populations.

Response: The key findings involve description in words. Instead of schematic illustration, we believe it is better to present them in Table 1. We have summarized the key findings and pointed out them for each study.

Comment 3: Changing the title of 2.3.2 to inducible and reporter models. Genetic-diet may get confused with the diet induced systems presented earlier in the review.

Response: We have changed the title accordingly.